# IL-6 Expression and the Confidence Interval-Based Estimation of Relevance (CIBER) Help Identify Persistent Inflammation and Cognitive Parameters of Executive Dysfunction in the Withdrawal Phase of Male Polydrug Abusers

**DOI:** 10.3390/healthcare13121462

**Published:** 2025-06-18

**Authors:** Jesua Guzmán-González, Alma Galvez-Contreras, Israel Jimenez-Navarro, Iris Perez-Alcaraz, Oscar Gonzalez-Perez, Rocio E. Gonzalez-Castañeda

**Affiliations:** 1Departamento de Psicología Básica, Centro Universitario de Ciencias de la Salud, Universidad de Guadalajara, Sierra Mojada n.º 950, Guadalajara 44100, Jalisco, Mexico; jesua.guzman@academicos.udg.mx; 2Departamento de Neurociencias, Centro Universitario de Ciencias de la Salud, Unidad de Atención en Neurociencias, Calle Centro Médico s/n Guadalajara 44340, Jalisco, Mexico; alma.galvez@academicos.udg.mx; 3Centro de Atención Integral en Salud Mental Estancia Prolongada (CAISAME E.P.), Antigua Carretera a Chapala, Guadalajara 45640, Jalisco, Mexico; 4Fundación Mexico me Necesita A.C., Libramiento sur Carretera Cajititlán 296, Cajititlán 45670, Jalisco, Mexico; iris.perez9926@alumnos.udg.mx; 5Maestría en Neurociencias de las Adicciones, Departamento de Neurociencias, Centro Universitario de Ciencias de la Salud, Sierra Mojada n.º 950, Guadalajara 44340, Jalisco, Mexico; 6Laboratorio de Neurociencias, Facultad de Psicología, Universidad de Colima, Avenida Universidad 333, Colima 28040, Colima, Mexico; osglez@ucol.mx

**Keywords:** CIBER, executive dysfunction, biomarker, polydrug consumers, addiction

## Abstract

**Background/Objectives**: Individuals diagnosed with substance use disorders (SUD) exhibit notable deficits in executive function (EFs). Notably, the pro-inflammatory cytokine interleukin-6 (IL-6) has been associated with cognitive impairments in individuals with substance use disorders. The specific neuropsychological parameters most affected by executive dysfunction remain poorly understood. **Methods**: In this study, sixteen patients diagnosed with SUD in the withdrawal phase were compared to twenty age-matched control subjects to ascertain which aspects of EFs were most adversely impacted. Plasma levels of IL-6 were quantified using enzyme-linked immunosorbent assay (ELISA). Data were analyzed using the Confidence Interval-Based Estimation of Relevance (CIBER) model to determine the most sensitive executive performance indicators. **Results**: Findings from the CIBER analysis revealed that the Wisconsin Card Sorting Test yielded the most pronounced cognitive discrepancies between males with and without SUD diagnoses. Elevated levels of IL-6 and associated executive dysfunction were observed to persist in males with SUD throughout the withdrawal phase. **Conclusions**: Notably, cognitive flexibility emerged as the most sensitive parameter indicative of executive dysfunction, suggesting its potential utility in tailoring clinical interventions for SUD patients during this critical recovery period.

## 1. Introduction

The Diagnostic and Statistical Manual of Mental Disorders, Fifth Edition (DSM-5) [1] defines substance use disorders (SUD) as cycles of consumption and relapses leading to pervasive substance abuse [2,3]. This cycle encompasses three distinct phases: the binge/intoxication phase, the withdrawal/negative affect phase, and the preoccupation/anticipation (craving) phase [4]. Individuals diagnosed with SUD commonly exhibit profound neuropsychological deficits, specifically in executive functions (EFs) [5]. EFs are critical cognitive processes facilitating independent, goal-oriented, self-directed, and adaptive behavior [6,7]. Miyake et al. [8] conceptualize them as comprising three core components: working memory, inhibition, and cognitive flexibility [8]. In patients with SUD, dysfunctions in these executive domains partially account for maladaptive consumption behaviors. They are linked to poor treatment adherence, high dropout rates [9], diminished motivation for change [10], and limited self-awareness [11], which are associated with higher rates of relapses that hinder rehabilitation [12].

A potential mechanism underlying executive dysfunctions in drug consumers is the inflammatory response mediated by activated microglia cells [13,14]. Reactive microglia secrete pro-inflammatory cytokines such as interleukin- 1 (IL-1), IL-6, and tumor necrosis factor-alpha (TNF-α) [15,16]. These pro-inflammatory cytokines can interact with monoaminergic terminals [17], potentially resulting in reduced activation of the ventral striatum and impaired cortico-striatal connectivity [18]. In clinical populations, elevated levels of IL-6 are correlated with reduced connectivity within the cortico-striatal-limbic regions [19], which supports the hypothesis that neuroinflammation contributes to hypoactivity in dopaminergic pathways and, consequently, executive function impairments through diminished activity in the frontal lobes [20,21,22].

Plasma levels of IL-6 positively correlate with increases in adrenocorticotropic hormone (ACTH), a key stress response regulator [23,24]. This relationship suggests a close association between stress and IL-6 levels [23]. Notably, the interaction between stress and IL-6 appears to be influenced by sex differences. Under physiological conditions, adult women exhibit elevated IL-6 levels in response to stressors compared to their male counterparts [25]. In clinical studies females with a history of amphetamine addiction, increased ACTH levels, potentially driven by estrogenic modulation, coincide with elevated IL-6 levels and corresponding impairments in cognitive flexibility [26,27]. Interestingly, research suggests that females may engage more rapidly in drug-seeking behaviors and cravings [28]. In contrast, males tend to experience more frequent issues related to substance intoxication and exhibit poorer rehabilitation outcomes [29]. Therefore, it is crucial to investigate whether males with SUD also show alterations in IL-6 levels and to explore the implications of these fluctuations on executive function performance during periods of abstinence.

The Confidence Interval-Based Estimation of Relevance (CIBER) is an innovative statistical framework introduced by Crutzen et al. [30] that facilitates the transformation of raw data into a visual representation, enabling the comparison of correlation coefficients, means, and confidence intervals. In this visualization, data are depicted as diamond shapes, which assist in evaluating effect sizes within meta-analyses employing univariate distributions [31]. In the present study, we applied this model to generate a comprehensive map illustrating multiple statistical comparisons among cognitive parameters encompassing working memory, inhibition, and cognitive flexibility performance in polydrug male consumers who have maintained 5.5 months of abstinence.

## 2. Materials and Methods

### 2.1. Participants and Recruitment

A cross-sectional, non-experimental observational study was conducted. Thirty-six male voluntary participants with an average age of 24 years were included and distributed in two groups. The first group consisted of 16 patients with SUD diagnosis who received a recent intervention scheme with a multimodal treatment for drug abuse disorders in the foundation “México Me Necesita” (Guadalajara City, México). In the group formed by participants from the foundation, the initial drug of choice was methamphetamine, which could be followed by inhalants and benzodiazepines. However, the exact doses are unknown because the subjects were unable to provide them in terms of weight. All the participants were from lower middle-income backgrounds (less than 4515 USD/year; World Bank). The second group was assembled with 20 healthy subjects paired by age, education and no history of drug consumption. The inclusion and exclusion criteria of SUD participants were based on individual case records. Inclusion criteria were: (1) male patients (18–45 years old) with a previous diagnosis and who voluntarily want to receive a resident scheme of rehabilitation in the foundation “México Me Necesita” and (2) at least 12 years of schooling. Exclusion criteria were: (1) a previous history of neurosurgical procedures, previous clinical history for head trauma, psychiatric disorders, and any visual condition that was not corrected by glasses, and voluntary withdrawal from the study. During the experimental design, the SUD participant did not suffer relapses. In the control group, we applied a clinical interview to identify past or present problematic consumption behavior. The subjects in this group reported a social consumption of alcohol, such as a couple of glasses of wine or beer cans at parties, tobacco (one or two cigars per week), and they denied the use of vape or consumption of any other licit or illicit drug. They also denied the presence of problems in their social, familiar, and academic/workplace environments related to alcohol or smoke consumption. In this group, we request to halt any consumption of alcohol or tobacco for at least one week before neuropsychological evaluation and blood testing. All the participants were informed about the specific aims or purposes of the study and were asked to voluntarily fill out and sign a consent letter. This protocol and its procedures were approved and supervised by the Ethical Committee (207/2021) of the Instituto Jaliscience de Salud Mental (SALME), Mexico, following the federal regulations and guidelines stipulated in the Mexican Official Norm (NOM-028-SSA2-1999 and NOM-012-SSA3-2012) and conducted following Declaration of Helsinki.

### 2.2. IL-6 Determination

To determine the levels of IL-6, peripheral blood of all participants was obtained between 8:00 and 9:00 a.m. Plasma was then obtained by centrifugation and stored at −80 °C until analysis. IL-6 plasma levels were determined with the kit Enzyme Immunoassay for IL-6 (Abcam, ab46042, Washington, DC, USA), following the step-by-step protocol provided by the manufacturer. All samples were analyzed by duplicate. Optical density was determined with a Multiskan FC model plate reader (Thermo Scientific Cat. 51119100, Waltham, MA, USA, EE.UU.) at 450 nm. The concentrations of each enzyme were expressed in pg/mL, and final concentrations were obtained by interpolation with the standard curve.

### 2.3. Neuropsychological Evaluation

The neuropsychological assessment was performed with the PsyToolkit stimulator (Inglaterra, Colchester) [31] visualized in a Chrome browser version 90.0.4430.212. To identify changes in Working memory (WM) tasks, the N-Back (NB) paradigm was set at level 2 of difficulty, and a variant with 15 letters from “A” to “T” was used in a total of 50 random trials on a black background screen. To confirm that the participant adequately understood the task, we employed two phases, one of the 1-Back and 2-Back training trials and the actual test, which starts directly in the 2-Back difficulty. The stimuli were presented at a rate of 250 ms with a 500-millisecond response limit. Feedback was provided through two colored strips above and below each stimulus: green for the correct answers and red for the wrong ones.

We used the classical rules and settings of the Go/No-Go paradigm to evaluate inhibitory control and attention. The participant must emit a response according to the stimulus presented; the green color indicates that they must press the button, while the red color indicates that they must not press it. Stimuli were displayed at a rate of 500 ms with a maximum response delay of 2000 ms. Two errors were quantified: pressing the button when a red “No-Go” stimulus appears, which indicates a failure to inhibit the response (commission error), and not pressing the button when a green “Go” stimulus appears, which indicates a failure to execute the required response (omission error).

Finally, we applied the Wisconsin Card Sorting Test (WCST) to identify cognitive flexibility. The participants had to sort cards under three possible options: shape, number, or color. A card that can be grouped with the three possibilities was presented every 250 ms; the subject had 5000 ms to match it with the possible options; if there was an error in the answer, auditory feedback with the word “no” was provided. The elements considered during the evaluation are perseverative and non-perseverative errors, correct answers, and response times [31].

### 2.4. Data Analysis

To identify the sampling distribution of the data, Shapiro-Wilk test was used. This analysis reported a (w = 0.589, *p* = 0.001), which signaled that the sample does not follow a normal distribution. In consequence, we used the Mann-Whitney U test to verify the effectiveness of the matching process for the sociodemographic data of age and schooling. The analysis did not find significant differences in age (U = 108.5, δ = 0.48, *p* = 0.15) and schooling (U = 186.5, δ = 0.27, *p* = 0.29) between the groups, which supports that both groups were properly matched. We also calculated the effect size for our findings, providing context for the observed differences, which helped to establish the practical significance of our results alongside *p*-values. To ensure a robust experimental design for detecting group differences, we established the statistical power for bidirectional independent samples: β = 0.80 and α = 0.05, with group sizes of n1 = 16 and n2 = 20. This configuration resulted in a minimum effect size (Cohen’s d) of 0.96 (r = 0.43) for Type-II error and 0.70 (r = 0.33) for Type-I error according to Lakens [32]. The Shapiro-Wilk test was employed to assess the sample distribution, which indicated a non-parametric distribution. Hence, the U Mann-Whitney was calculated to determine differences between groups. We apply Spearman correlation (ρ) to identify the association between IL-6 levels and cognitive parameters. The CIBER method generates a visual map with two panels. In the left panel, the diamonds represent the 99.99% confidence interval (CI) for each cognitive parameter, and the diamonds’ color represents the groups. In the right panel, the diamonds represent the mean of each cognitive item, and the more intense the color of the diamond, the greater the meaning of each item (strength association), whereas the less intense the color of the diamond, the lower the item’s relevance. Thus, the redder the diamond, the stronger the negative association, whereas the greener the diamond, the stronger the positive association. The lines in the right panel are graduated according to the standard deviation (SD) from −2 to +2 with 95% CIs. The quantitative analysis was expressed as 95% CIs (lower limit/upper limit). To enhance the CIBER analysis, a multivariate analysis of covariance (MANCOVA) was conducted. The dependent variables were the results of the neuropsychological tests, with the group (control or SUD) as the fixed factor. As our analysis involved a single root and evaluated all discriminant functions, Pillai’s trace test was applied. Results were reported on a scale from 0 to 1, where higher values indicate a better fit, alongside degrees of freedom and significance at an alpha level of <0.05 with 10 degrees of freedom.

## 3. Results

### 3.1. The Plasma Levels of IL-6 Are Increased in Polydrug Male Consumers During Abstinence

The participants in the SUD group were polydrug consumers of methamphetamines, tobacco, and alcohol. In this group, 31.58% also had a consumption of cannabinoids, 10.53% inhalants, and 5.26% benzodiazepines (Table 1), with a range of abstinence time between 100–324 days. We quantified the plasmatic IL-6 and found that the SUD group had high levels of this cytokine, 13.16 pg/mL (IQR = 12.07–14; n = 16 subjects) when compared to the control group 10.04 pg/mL (IQR = 8.64–14.10; n = 20 subjects) and these differences were statistically significant (U = 93, *p* = 0.033) (Figure 1). Tukey’s method was used to assess potential outliers in IL-6 levels. The elevated value was not identified as an extreme outlier and was retained due to its clinical relevance (t = 2.09, *p*_tukey_ = 0.044).

### 3.2. Executive Dysfunction Persists in Polydrug Males During Abstinence

We observed a lower EFs’ performance in the SUD group (Table 2). In the evaluation of working memory performance with the N-Back cognitive parameter, the SUD group had the worst performance (80.00 correct answers, IQR = 74.67–88.00; n = 16) in comparison to the control group (88.66 correct answers, IQR = 82.33–92.00; n = 20. U = 86, *p* = 0.019). The SUD group also shows more reaction times (25.96, IQR = 24.17–26.66; n = 16 subjects) vs the control group (25.00, IQR = 22.36–25.59; n = 20 subjects. U = 94, *p* = 0.037). This evidence indicates that patients present poor working memory performance and require more time to monitor and update the information.

In the Go/No-Go task, we measured the inhibitory capacity through the results of the correct answers in the Go condition and found that the SUD group present a poor inhibitory capacity (100, IQR = 97.50–100; n = 16 subjects) when compared to the control group who present a median of 100 in this score (IQR = 100–100; n = 20 subjects. U = 110, *p* = 0.009). Regarding the reaction time, the SUD group responded quicker (37.56, IQR = 36.50–44.03; n = 16 subjects) than the control subjects (43.91, IQR = 38.39–48.13; n = 20 subjects. U = 88, *p* = 0.023), whereas commission in the No-Go condition, the control group obtained higher correct answers (100, IQR = 97.50–100; n = 20 subjects) as compared to the SUD group (90.00, IQR = 80.00–90.00; n = 16 subjects. U = 58.5, *p* = 0.001). Furthermore, the reaction time in the SUD group was longer (18.27, IQR = 16.49–18.31; n = 16 subjects) than in controls (=20, IQR = 19.59–20.00; n = 20 subjects. U = 60, *p* = 0.001). Altogether, these results indicate that the ability to inhibit responses and verify the type of response is significantly deficient in the SUD group.

We assessed cognitive flexibility with the WCST task and found that the SUD group had a significantly worse performance in correct answers (65.83, IQR = 47.08–73.75; n = 16 subjects) as compared to the control group (81.67, IQR = 78.33–83.33; n = subjects. U = 35, *p* = 0.001). We do not find significant differences in reaction times between the SUD (20.59, IQR = 15.36–24.60; n = 16 subjects) and the control group (15.82, IQR = 13.65–17.96; n = 20 subjects. U = 0.101, *p* = 0.063). In perseverative errors, we found statistical significance (U = 53.5, *p* = 0.010) between the SUD (20.00, IQR = 15.00–27.08; n = 16 subjects) and control group (11.67, IQR = 10–15; n = 20 subjects), which mean that SUD group had more significant difficulties in changing its responses when faced with different forms of classifications. In contrast, the non-perseverative errors of SUD subjects (10.00, IQR = 6.25–19.17; n = 16 subjects) were also higher than in controls (6.67, IQR = 5.00–8.75; n = 20 subjects), but without statistical significance (U = 99, *p* = 0.053). Altogether, these data indicate that individuals with SUD have significantly impaired cognitive flexibility compared to controls, shown by fewer correct answers and more perseverative errors in the WCST. While reaction times did not differ significantly, higher perseverative errors indicate difficulty adapting to new rules. Increases in non-perseverative errors also suggest broader cognitive flexibility challenges.

### 3.3. Correlations Between Cognitive Task Performance and Reaction Time

To identify the strength of the association between correct answers and reaction time across all cognitive tasks, we conducted a correlation analysis via the Spearman test (Table 3) and found a moderate negative association between the reaction times and right answers in the N-Back test (Spearman’s rho of −0.538, *p* = 0.001). This indicates that participants who perform more efficiently in terms of correct responses also tend to have faster reaction times. This pattern may imply that the ability to maintain information actively in working memory is related to faster processing of that information.

In the Go/No-Go task, a significant moderate positive correlation was found between right answers of Go and No-Go conditions (Spearman’s rho = 0.476, *p* = 0.003), which suggests that participants who perform well in the “go” section also tend to perform well in the “No-go” section. This could reflect that good performance in one executive function (such as controlled impulsivity in the “go” section) is associated with adequate inhibitory control in the other (inhibiting responses in the “no-go” section). The association between the parameters of right answers and reaction times indicates a low relationship (Spearman’s rho = 0.151, *p* = 0.38), which suggests that accuracy and speed in this task may be functioning relatively independently. Specifically, a participant may be accurate in their responses but take more time to do so, or they may respond quickly but make more errors, suggesting that response control and speed are not strongly linked or reflect different cognitive processing strategies. The analysis of right answers ind Go condition and reaction times in No Go condition results in a moderate positive relationship between these two variables (Spearman’s rho = 0.515, *p* = 0.001), which could suggest an interaction between response control and the speed of inhibition. Specifically, participants who are more accurate in the “go” section seem to take more time to inhibit their responses in the “no-go” section.

In the Wisconsin Card Sorting Test, several strong negative correlations were identified. Right answers correlated negatively with reaction times in WCST, showing a moderate correlation (Spearman’s rho = −0.564, *p* = 0.001). This indicates that those with faster reaction times tend to perform better on the task, reflecting greater cognitive efficiency and quicker mental flexibility, which are traits of individuals with stronger executive functions. The high correlation between right answers and perseverative errors in WCST (Spearman’s rho = −0.790, *p* = 0.001) strongly suggests that participants who respond correctly on the test make fewer perseverative errors. Since perseverative errors are a key measure of cognitive flexibility, a higher number of correct responses is linked to a better ability to shift strategies or adapt to new rules without relying on previously incorrect rules. This relationship implies that those with stronger executive control and greater cognitive flexibility (fewer perseverative errors) are more effective in the task. Regarding the association between right answers and non perseverative errors in WCST (Spearman’s rho = −0.760, *p* = 0.001), the data suggests that participants who provide correct answers also make fewer non-perseverative errors. These results strongly indicate that those performing better on the task are also more proficient at adapting to rule changes without making errors related to the incorrect application of new strategies.

There was also a significant positive and moderate correlation between reaction times and perseverative errors in WCST (Spearman’s rho = 0.536, *p* = 0.001), suggesting that slower response times may be linked to difficulties in adapting to new rules or cognitive flexibility. Alternatively, longer reaction times might indicate doubt when shifting strategies, leading to persistent incorrect responses or an inability to adapt appropriately to rule changes. Regarding the relationship between reaction times and non-perseverative errors in WCST (Spearman’s rho = 0.533, *p* = 0.001), it reflects that individuals who have longer reaction times tend to make more non-perseverative errors. This data suggests a difficulty in applying new rules, possibly due to slower decision-making or an inability to inhibit incorrect responses.

### 3.4. Plasma IL-6 Levels Correlate to Working Memory Performance in Adult Males

To identify if IL-6 levels are associated with working memory, inhibition, and cognitive flexibility performance. In the analysis of all groups, the plasmatic IL-6 levels present a negative correlation with the number of correct answers (r = −0.397; *p* = 0.017) and a positive correlation with reaction time (r = 0.347; *p* = 0.038) in the N-Back test whereas in the Go/No-Go task, the cytokine levels show a negative correlation with the number of correct answers in the Go condition (r = −0.321; *p* = 0.05). The other cognitive parameters evaluated did not show statistical differences (Table 2). When we compared the association between IL-6 levels with FE performance per group, we found that in the control group, the plasma IL-6 levels only presented a positive correlation with the reaction time in the N-Back test (r = 0.572; *p* = 0.008), whereas, in the SUD group, the plasmatic IL-6 levels did not show statistical significance, except by a limitrophe statistical significance in the number of perseverative errors (r = 0.472; *p* = 0.065).

### 3.5. Inhibition and Cognitive Flexibility Are the Most Sensitive Cognitive Parameters That Differentiate EFs’ Performance Between the Groups

We applied the CIBER method to obtain a map and analyze the distance and data distribution of different cognitive parameters that comprise working memory, inhibition, and cognitive flexibility performance. In the left panel, we described the cognitive parameters as follows. For working memory, NBack correct answer (NBra) and NBack reaction time (NBrt). For inhibition, Go-no-go correct answer Go (Gora), Go-no-go right answer no-go (NoGora), Go-no-go reaction time Go (Gort), Go-no-go reaction time no-go (NoGort). For cognitive flexibility, WCST correct answer (WCSTra), WCST reaction time (WCSTrt), WCST perseverative error (WCSTpe), and WCST non-perseverative error (WCSTnpe) (Figure 2).

Visually, the left panel represents the distance in performance of each cognitive parameter according to the group. In this panel, diamond shapes represent 99.99% CI, and diamond colors indicate the group. The quantitative analysis was carried out by identifying the CIs (lower limit/upper limit) and the overlap index to obtain mean differences between groups of each cognitive parameter. For working memory, NBra 6.37 (1.47|11.26), NBrt 1.33 (0.06|2.60). For inhibition, Gora 0.78 (0.24|1.32), NoGora 11.75 (4.48|19.02), Gort 6.01 (.92|11.08), and NoGort: 2.01 (0.75|3.26). For cognitive flexibility, WCSTra of 19.41 (11.26|27.6), WCSTrt 5.15 (1.3|9), WCSTpe 8.3 (3.51|13.06), and WCSTpe 8.3 (1.43|15.3). According to the analysis of the data in the left panel, we can observe that NoGora for inhibition, WCSTra and WCSTpe for cognitive flexibility are the most sensitive parameters for identifying cognitive impairment in SUD patients during the withdrawal phase. This sensitivity suggests these measures could be particularly effective in neuropsychological assessments and interventions for these individuals (Figure 2).

The analysis in the right panel (Figure 2) shows the delta (δ) values representing the mean differences between SUD and control groups across cognitive parameters, with associated 95% CIs (lower limit/upper limit). In this right panel, the lines are graduated according to 95% CIs from −2 to +2 and indicate the direction of the association of each cognitive parameter. Therefore, the redder and greener the diamond, the stronger the directionality of the association, whereas the grayer (duller) the diamond, the weaker the association. Thus, NoGora (δ = 1.101 (0.3463|1.8337), WCSTra (δ = 1.623 (0.7759|2.4428), and WCSTpe (δ = 0.822 (0.106|1.5196) were the cognitive parameters with higher distance data distribution. Our data indicated that NBra δ = 0.886 (0.1621|1.5911) and NBrt a δ = 0.716 (0.0128|1.4033) suggested moderate differences in working memory performance, indicating that individuals in the SUD group typically perform worse than controls. In the evaluation of inhibition, we obtained the following data Gora δ = 0.983 (0.2453|1.6992), NoGora δ = 1.101 (0.3463|1.8337); Gort δ = 0.805 (0.0912|1.5009) and NoGort δ = 1.089 (0.3359|1.8197). These findings indicate significant differences in inhibition performance, especially in the “no-go” task, which reflects challenges in impulse control of SUD individuals. When analyzing cognitive flexibility, we found a marked impact on this ability in the SUD group when compared to controls, WCSTra δ = 1.623 (0.7759|2.4428) and WCSTrt a δ = 0.914 (0.1857|1.6216), which was confirmed with WCSTpe a δ = 1.184(0.4158|1.9282) and WCSTpe a δ = 0.822 (0.106|1.5196) parameters. Altogether, this evidence indicates that the number of right answers in the No Go condition for inhibition response and the number of correct answers plus perseverative errors in the Wisconsin test for flexibility may be potential cognitive targets to be addressed in neuropsychological treatments in SUD patients and monitoring their therapeutic progress. Hence, scientific evidence suggests that cognitive parameters could function as a prediction of relapses in addictions [33,34], a deep statistical analysis would be necessary to evaluate the impact of cognition in drug recovery. To strengthen the CIBER analysis, we apply a multivariate analysis to the data (Table 4). The statistical significance obtained in them points out the significance of the CIBER model to identify the relationship between IL-6 and cognitive parameters measured (Pillai’s Trace = 0.77, F = 8.69, *p* ≤ 0.001).

## 4. Discussion

In the present study, we report that polydrug consumers show a persistent increase in IL-6 plasma levels and deficits in working memory, inhibition, and cognitive flexibility performance during the withdrawal phase. Interestingly, IL-6 levels have a negative correlation with the number of correct answers and a positive correlation in the reaction time in the N-Back test for working memory. The CIBER analysis indicated that the inhibition response (correct answers in the No Go condition) and cognitive flexibility (correct answers and perseverative errors) are parameters to assess cognitive performance between the groups. Whereas the intercorrelations between reaction time and right answers in the tasks show a moderate association.

The presence of higher levels of interleukins has been demonstrated in drug addiction. In a recent meta-analysis by Wei [35], they report that individuals with SUD exhibit a dysregulated immune profile, characterized by elevated peripheral levels of both anti-inflammatory cytokines (IL-10 and IL-4) and pro-inflammatory cytokines (TNF-α, IL-8, and IL-6). The findings suggest that chronic substance use, including methamphetamine, may trigger systemic inflammation through mechanisms involving oxidative stress, gut permeability, and blood-brain barrier disruption. In this sense, it is important to mention that peripheral levels of IL-6 through different mechanisms, such as active transport across the blood-brain barrier and vagal nerve-mediated signaling. There are evidence were peripheral cytokines such IL-6 may influence central nervous system function, inducing reactive gliosis, and promoting neuroinflammatory cascades that impair synaptic plasticity and cognitive performance. In polydrug consumers, chronic systemic inflammation and cytokine dysregulation may exacerbate these effects, and maybe leading to greater cognitive impairments [36].

Others biological mechanisms related to immune dysregulation includes overstimulation of the HPA axis [37] followed by an increase of glucocorticoids [38]. Recently, it has been proposed that in addictive behavior, exist biological and cultural intersex differences [39], which may affect the development, maintenance, and treatment response [40]. At the biological level, these sex differences are related to changes in hypothalamic-pituitary-adrenal (HPA) axis activity and mesolimbic sensitivity to dopamine [41]. Specifically, females present a decrease in the expression of D3 receptors in the nucleus accumbens [42] and higher levels of two biomarkers of stress, cortisol, and ACTH [23,43], when compared to males. Interestingly, there is co-stimulation between the increase of ACTH and IL-6 levels [23], which suggests that a stressor stimulus can increase the hyperactivity of HPA and maintain higher IL-6. Specifically, has been reported that IL-6 stimulates HPA axis that in turn enhances drug-related learning and increases the susceptibility to drug use or craving via alter neuronal activity in the dopaminergic synthesis nucleus, the ventral tegmental area, and modifies the neurotransmitter of GABA, glutamate and dopamine [44]. Although the availability of D2 receptors in the orbitofrontal cortex has also been involved in IL-6 dysregulation [45]. D2 receptors have been suggested to regulate the connection between the immune and nervous systems [46].

Both cocaine and methamphetamine can alter the dopaminergic system via Toll-like receptors (TLRs) [47] which are expressed in glial cells and have been related to neuroinflammation [48]. Specifically, the TLR4 receptor not only regulates the neuroinflammatory response but also the rewarding effects of cocaine abuse [47]. In preclinical studies of methamphetamine abuse, Naloxone, a TLR4 inhibitor, reduces the dopamine levels in the nucleus accumbens (Namba et al., 2021 [49]). At respect, several dopamine agonist [50] and immunomodulatory treatments have been proposed [47,51] as a pharmacological strategies to target IL-6 related inflammation or behavioral effects of drug abuse [52,53]. However, a challenge of these treatments is that drug abuse generates diverse effects in the brain and involves several biological mechanisms [51] that require a set of therapeutic strategies. In female drug abusers, high levels of Il-6 tend to persist for three months after drug consumption cessation [27]. In our study, we found that IL-6 can remain increased for at least 5.5 months during the abstinence stage in males.

In a recent meta-analysis, it has been reported that the normal range of IL-6 plasma levels in healthy individuals is 0 pg/mL to a maximum of 43.5 pg/mL, with a typical average range between 0.13 and 30.46 pg/mL [49]. In our study, we observed that the expression levels in the SUD group are higher compared to the control group, yet they fall within the range reported by this meta-analysis. Other evidence found values similar to ours, specifically higher IL-6 levels in amphetamine users compared to their controls, which also fall within the same range reported by this meta-analysis [54]. Although our data fall within the normal range, the elevated levels of IL-6 in individuals with SUD highlight the necessity for clinical awareness and further research to define their potential implications in clinical management.

IL-6 is considered one of the pivotal inflammation markers related to executive functions such as working memory [55]. Recently, the CIBER analysis was proposed as an innovative statistic model focused on identifying the social and cognitive determinants [56]. It also evaluates intention and self-efficacy to assist interventions in pregnant adolescents [57], as well as beliefs and behaviors of health providers [58], obesity [59], and more. However, in recent years, this model has also been proposed as an effective tool to identify cognitive performance in pathophysiological conditions such as alcoholism [60]. Our results indicate that the performance in inhibition and cognitive flexibility showed the highest score difference among all the executive functions evaluated, suggesting that they could be pivotal targets for designing neuropsychological rehabilitation programs during abstinence treatments in polydrug consumers. Identifying the cognitive parameters that need improvement in neuropsychological treatments is quite relevant because EFs’ performance is a strong predictor of treatment adherence during abstinence [61].

Cognitive flexibility is a process that regulates goal-directed behavior and is part of the mental set-shifting ability [62]. WCST is a reliable test to evaluate set-shifting. Hence, this test has been proposed as a cognitive indicator of frontal dysfunction. Specifically, a high number of perseverative errors (an error that occurs when a subject organizes or classifies a card according to a pre-established rule and not according to the current one) has been associated with lateral or dorsomedial prefrontal lesions [63]. In subjects with remitted smoking addiction, brain lesions in the insula, dorsal cingulated cortex, and dorsolateral prefrontal cortex have been reported, and a similar pattern of lesions has been suggested with other drug addictions [64]. Colzato et al. [65] report that the recreational use of stimulant drugs such as cocaine plus alcohol could be associated with increases in perseverative errors via dysfunction of DA2 receptors because of pint-size ischemic strokes [66]. In this paper, our analyses with the CIBER model identify perseverative errors as the most sensitive cognitive parameter, which may facilitate neuropsychological diagnosis, and rehabilitation plans in these patients.

Cognitive flexibility requires the coordination and interaction between updating (working memory) and inhibition responses [62]. In this regard, the Go/No-Go test is one of the primary assays used to investigate inhibition responses [67], which evaluates the ability to respond to stimuli (go condition) while discriminating and stopping other stimuli (no-go condition) [68]. The score in the no-go condition indicates commissions, a type of error based on premature responses related to impulsivity [66]. In drug abusers, it has been suggested that failures in inhibition response underlie the impulsivity behavior that favors consumption [69]. According to the CIBER analysis, both correct answers in the no-go condition (commissions) and perseverative errors are highly relevant cognitive parameters to be evaluated during the withdrawal phase of drug addiction. This information will help clinicians design neuropsychological assessments and programs that will improve inhibition response since cognitive function has been reported as a predictive parameter of relapse in some kinds of drugs [33,34].

In our study, we found that a worse performance in executive function persisted for at least 5.5 months during the abstinence stage. According to McGregor et al. [70], the natural course of withdrawal in methamphetamine patients occurs in two phases, an acute one that involves the first 24 h to 7 days following cessation. This phase comprises a linear decrease in the severity of the symptoms and a subacute phase where the symptoms remain relatively stable (the next two weeks). In the same sense, Zoric et al. in [71] reported similar findings and signaling that physiological symptoms decrease in the first weeks, but psychological symptoms as craving, could be present until six months after cessation. Depending on the type of drug abuse, the abstinence period is associated with sustained cognitive impairments. For example, a lack of recovery in cognitive functioning following the cessation of benzodiazepine has also been reported [72], while Loeber et al. showed that patients with alcohol abuse who underwent 3 months of withdrawal display poorer performance in executive function [73] that persists beyond 6 months [74] compared to healthy controls. Other evidences show that methamphetamine abusers have partial improvement over the course of 6 months [75].

Our data shows no significant age difference between the SUD group, with all participants aged in their 20 s to 30 s. Since age can affect cognitive performance, especially reaction times, previous studies indicate that age-related differences are more prominent with a larger age gap of around 30 years [76]. In our study, with all subjects within a similar age range, we believe age does not account for the differences observed. Similarly, reaction time in the N-Back task and Go-No-Go [77].

While our study offers significant insights, it is essential to acknowledge certain limitations. The relatively small sample size for group comparisons might have influenced the depth of the analysis. Although the CIBER model is an approach aimed at reducing data uncertainty, it should be evaluated alongside other clinical factors, such as medical history and individual responses to treatments. Future research could enhance the robustness of network analyses by expanding the SUD sample size, facilitating progress from theoretical to causal network interpretations. Additionally, prospective studies should focus on longitudinally monitoring SUD patients to explore potential correlations between relapse frequency and duration, interleukin-6 (IL-6) levels, and executive function performance. These approaches will further enrich our understanding of their long-term effects.

One limitation of the present study is that matching was based solely on age and education, without accounting for other potentially influential factors such as socioeconomic status, smoking, or other lifestyle variables. While this may have introduced some uncontrolled variability, the main aim was to explore broad associations between executive function, inflammation, and polydrug use. Moreover, although the types of substances used were recorded, detailed quantitative data (e.g., frequency, dose, duration) were not available, which may limit the granularity of our interpretations. Nevertheless, these findings provide a valuable foundation for future research that includes more detailed assessments and a broader range of control variables.

The study findings were also hindered by a lack of in-depth exploration into the use of vaping devices among the analyzed population. However, given that patterns of psychostimulant use are constantly involving among young adults, addressing this variable in future research would be essential.

## 5. Conclusions

During the abstinence phase, polydrug users demonstrate marked deficits in working memory, inhibition, and cognitive flexibility, coupled with elevated plasma IL-6 levels. These neurocognitive impairments were evidenced by diminished correct responses in the Go/No-Go test and increased perseverative errors, suggesting that these may serve as cognitive biomarkers for diagnosing or monitoring the therapeutic progress in these patients. Targeting these cognitive deficits could guide the development of neuropsychological interventions such as self-impulse control training, tasks for adapting to new circumstances, strategies for skill development, and problem-solving and decision-making skills aimed at rehabilitating executive dysfunction in males with SUD during withdrawal, thereby enhancing treatment efficacy and reducing relapse risk.

## Figures and Tables

**Figure 1 healthcare-13-01462-f001:**
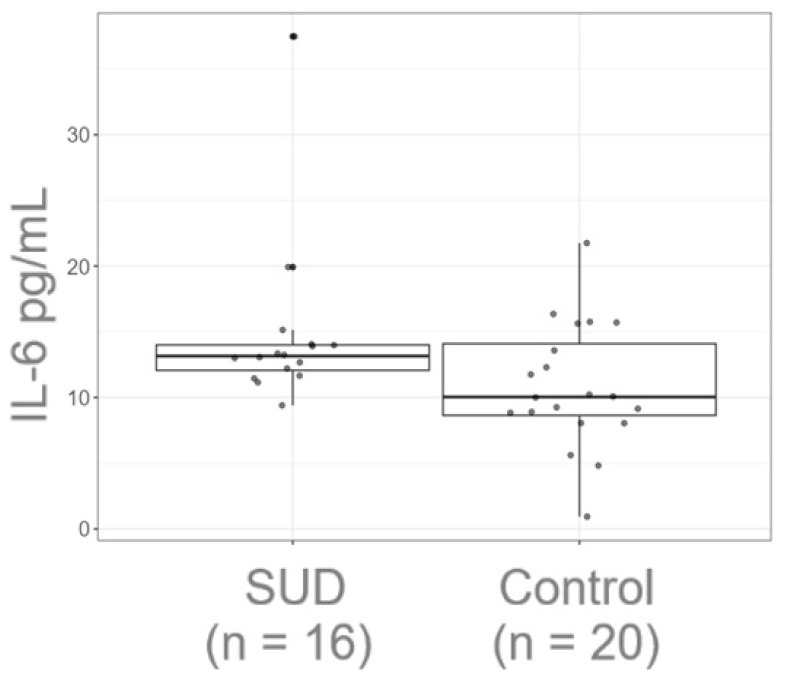
Peripheric IL-6 levels between groups. Boxplot of the peripheral levels of IL-6 and their differences between the groups (rbr = 0.42), the red dot symbolizes the median of the groups, which was for the SUD group 13.2 (12.07–14) and 10 (8.64–14.10) for the control group. The black points on the boxplot represents the individual data for each group.

**Figure 2 healthcare-13-01462-f002:**
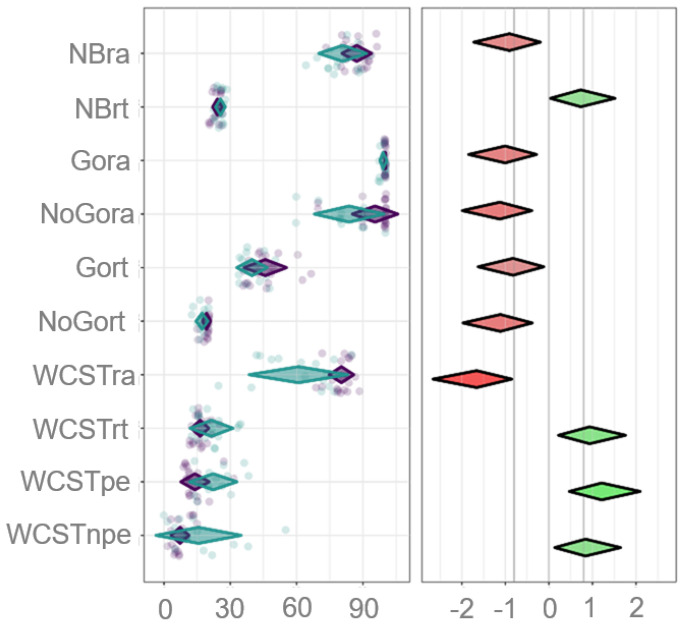
CIBER plot of EF performance visualizing means and differences between. Control and SUD groups. The fill color of the diamonds indicates the group: SUD (aquamarine) and control (purple). Each dot indicates the score of each subject and the distance between diamonds represents the difference of distribution data. In the right panel the diamond colors indicate the direction with green colored suggesting a positive association compared to control and red having a negative association compared to control. Additionally, the strength of difference is indicated by color intensity with stronger colors representing stronger associations.

**Table 1 healthcare-13-01462-t001:** Sociodemographic characteristics of the SUD group and control.

	Control Med (IQR)	SUD Med (IQR)	Statistical Test
Age	21.0 (2.0)	24.5 (7.75)	U = 108.5, δ = 0.48, *p* = 0.15
Year of schooling	12.0 (0.0)	12.0 (0.0)	U = 186.5, δ = 0.27, *p* = 0.29
Days with abstinence		171 (59.8)	
**Consumption type**		***f* (%)**	
Alcohol		16 (100%)	
Tobacco		16 (100%)	
Inhalants		2 (10.53%)	
Cannabinoids		6 (31.58%)	
Benzodiazepines		1 (5.26%)	
Methamphetamines		16 (100%)	

Sociodemographic data are expressed as the median (Med) and interquartile rank (IQR), and frequency (f) of. Statistical analysis: Mann-Whitney *U* test.

**Table 2 healthcare-13-01462-t002:** Performance of the SUD and control groups in the neuropsychological test.

Test	Control Med (IQR)	SUD Med (IQR)	*U*	*p*
NBack right answer	88.66 (82.33–92.00)	80 (74.67–88.00)	86	0.019 *
NBack reaction time	25.00 (22.36–25.59)	25.96 (24.17–26.66)	94	0.037 *
Go-no-go right answer Go	100 (100–100)	100 (97.50–100)	110	0.009 **
Go-no-go right answer no-go	100 (97.50–100)	90 (80.00–90.00)	58.5	0.001 *
Go-no-go reaction time Go	43.91 (38.39–48.13)	37–56 (36.50–44.03)	88	0.023 *
Go-no-go reaction time no-go	20.00 (19.59–20.00)	18.27(16.49–18.31)	60	0.001 ***
WCST right answer	81.66 (78.33–83.33)	65.83 (47.08–73.75)	35	0.001 ***
WCST reaction time	15.82 (13.65–17.96)	20.59 (15.36–24)	101	0.063
WCST perseverative errors	11.66 (10–15)	20 (15.00–27.08)	53.5	0.001 ***
WCST non-perseverative errors	6.66 (5.00–8.75)	10 (6.25–19.17)	99	0.053

Data are expressed as the median (M) and the interquartile rank (IQR). WSCT = Wisconsin sort card test; NB = N-Back test. Statistical differences were established with the Mann-Whitney *U* test (*p* < 0.05). *p* = <0.05 *, <0.01 **, <0.001 ***.

**Table 3 healthcare-13-01462-t003:** Correlations of interleukin-6 with neuropsychological performance by cognitive dimension.

Dimension	SUD	Control	Correlation
Interleukin 6	Impaired	Preserved	
Working memory	Impaired	Preserved	
Reaction times			+
Right Answers			−
Inhibitory control	Impaired	Preserved	
Reaction times			+
Right Answers			−
Cognitive flexibility	Impaired	Preserved	
Reaction times			−
Right Answers			+
Non perseverative errors			+
Perseverative errors			+

The table shows the direction of the correlation based on interleukin-6: an plus sign indicates a positive correlation, while a minus indicates a negative correlation.

**Table 4 healthcare-13-01462-t004:** Univariate Tests that support CIBER conclutions.

	Dependent Variable	Sum of Squares	df	Mean Square	F	*p*
Group	NBra	360.29	1	360.287	6.98	0.012
	NBrt	15.74	1	15.737	4.56	0.04
	Gora	5.43	1	5.425	8.59	0.006
	NoGora	1227.22	1	1227.222	10.78	0.002
	Gort	320.49	1	320.494	5.76	0.022
	NoGort	35.84	1	35.841	10.54	0.003
	WCSTra	3351.17	1	3351.173	23.4	<.001
	WCSTrt	236.16	1	236.156	7.42	0.01
	WCSTpe	611.13	1	611.127	12.46	0.001
	WCSTnpe	623.47	1	623.472	6.01	0.02
Residuals	NBra	1753.85	34	51.584		
	NBrt	117.28	34	3.449		
	Gora	21.48	34	0.632		
	NoGora	3870	34	113.824		
	Gort	1890.5	34	55.603		
	NoGort	115.58	34	3.4		
	WCSTra	4868.19	34	143.182		
	WCSTrt	1081.98	34	31.823		
	WCSTpe	1668.19	34	49.065		
	WCSTnpe	3528.61	34	103.783		

## Data Availability

The data is unavailable due to clinical privacy, but it can be requested from the authors of the article.

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
