# Peer review of "IL-6 Expression and the Confidence Interval-Based Estimation of Relevance (CIBER) Help Identify Persistent Inflammation and Cognitive Parameters of Executive Dysfunction in the Withdrawal Phase of Male Polydrug Abusers"

_healthcare, 2025, doi:10.3390/healthcare13121462_

Round 1
Reviewer 1 Report
Comments and Suggestions for Authors
Overall, this is a clear, concise, and well-written manuscript. The study highlights the role of IL-6 expression and CIBER analysis in detecting persistent inflammation and executive dysfunction during the withdrawal phase in male polydrug users. The authors have also appropriately acknowledged the limitations of the current study. However, the following suggestions may help improve the scientific content before publication:
- Minimize the use of abbreviations, as excessive abbreviation may disrupt the flow and readability of the manuscript.
- The authors have employed the N-Back paradigm, Go/No-Go paradigm, and the Wisconsin Card Sorting Test (WCST), which are appropriate for assessing working memory, inhibitory control, and cognitive flexibility, respectively. However, these tasks do not fully capture real-life decision-making or sustained attention during withdrawal in polydrug users. The authors may consider including the Continuous Performance Test to assess sustained attention, the Iowa Gambling Task for decision-making, and the Tower of London task to evaluate planning ability.
- The authors may also consider discussing potential therapeutic interventions that could be effective during the withdrawal phase, particularly those targeting inflammation and cognitive deficits.
Author Response
Guadalajara, México. May 19th 2025
Re: Manuscript’s resubmission (ID: healthcare-3565164)
MDPI Healthcare Editorial Office
Grosspeteranlage 5, 4052 Basel, Switzerland
Dear Editorial Office Team,
We are grateful for the review and comments from your editorial board and your consideration of our revised manuscript entitled “IL-6 expression and the Confidence Interval-Based Estimation of Relevance (CIBER) help identify persistent inflammation and cognitive parameters of executive dysfunction in the withdrawal phase of male polydrug abusers” for publication in the journal Healthcare.
In our previous version, the reviewers were positive about our manuscript and made several constructive suggestions, which we have followed to improve our manuscript. In this revised version, we have addressed the following suggestions:
- Better described the method, results, and discussion section
- Added bibliographic information throughout the manuscript
Sincerely,
Dr. Rocio Gonzalez-Castañeda on behalf of the authors.
Reviewer #1. Overall, this is a clear, concise, and well-written manuscript. The study highlights the role of IL-6 expression and CIBER analysis in detecting persistent inflammation and executive dysfunction during the withdrawal phase in male polydrug users. The authors have also appropriately acknowledged the limitations of the current study. However, the following suggestions may help improve the scientific content before publication:
Response: Thanks to reviewer #1 for their kind comments about the manuscript and their constructive suggestions.
Reviewer #1: Minimize the use of abbreviations, as excessive abbreviation may disrupt the flow and readability of the manuscript.
Response: Thanks to the reviewer for bringing this to our attention. In this revised version, we reduced the number of acronyms by 30% to improve the reading fluency. Please see lines 45, 56, 93, 186, 187, 189, 186, 193, 195-197, 197-199, 199, 205- 208, 209-211, 213, 214-216, 227, 232, 236, 241, 243, 249, 253-255, 258-261, 266-269, 271- 272, 279-283, 285, 289
Reviewer #1: The authors have employed the N-Back paradigm, Go/No-Go paradigm, and the Wisconsin Card Sorting Test (WCST), which are appropriate for assessing working memory, inhibitory control, and cognitive flexibility, respectively. However, these tasks do not fully capture real-life decision-making or sustained attention during withdrawal in polydrug users. The authors may consider including the Continuous Performance Test to assess sustained attention, the Iowa Gambling Task for decision-making, and the Tower of London task to evaluate planning ability.
Response: We appreciate the reviewer’s comment and agree that decision-making, as assessed by the Iowa Gambling Task, and sustained attention are relevant neuropsychological functions worth exploring. However, these processes fall outside the scope of the present study. Our work is grounded in the executive functioning model proposed by Miyake et al. (2000), which specifically identifies working memory, inhibitory control, and cognitive flexibility as core components. While we acknowledge the value of including additional assessments such as the Continuous Performance Test or the Tower of London in future research, these measures were not aligned with the original aims and methodological design of our study. Please see page 2, line 44-46.
Reviewer #1: The authors may also consider discussing potential therapeutic interventions that could be effective during the withdrawal phase, particularly those targeting inflammation and cognitive deficits.
Response: We appreciate the reviewer’s suggestion and agree that it is important to consider therapeutic interventions during the withdrawal phase, particularly those targeting inflammation and cognitive deficits. Recent research has highlighted immunopharmacology as a promising approach for the development of new pharmacotherapies that promote long-term abstinence and minimize the harmful effects of substance use disorder (SUD) (Namba et al., 2021). Accordingly, we have added this perspective to the discussion section, emphasizing the potential relevance of this mechanism for mitigating SUD-related comorbidities and improving treatment outcomes. Please see page 12, lines 463 to 471
References:
Miyake, A., Friedman, N. P., Emerson, M. J., Witzki, A. H., Howerter, A., & Wager, T. D. (2000). The Unity and Diversity of Executive Functions and Their Contributions to Complex “Frontal Lobe” Tasks: A Latent Variable Analysis. Cognitive Psychology, 41(1), 49–100. https://doi.org/10.1006/cogp.1999.0734
Namba, M. D., Leyrer-Jackson, J. M., Nagy, E. K., Olive, M. F., & Neisewander, J. L. (2021). Neuroimmune Mechanisms as Novel Treatment Targets for Substance Use Disorders and Associated Comorbidities. Frontiers in Neuroscience, 15. https://doi.org/10.3389/fnins.2021.650785
Reviewer 2 Report
Comments and Suggestions for Authors
The current study investigated the potential relationship between persistent inflammation and cognitive deficits in executive functions. Plasma IL-6 was used as a single marker for inflammation, and working memory, inhibitory control and attention, and cognitive flexibility were assessed using several neuropsychological tests during the withdrawal phase in male polydrug abusers compared to healthy controls. The study reported elevated IL-6 levels and executive dysfunction in the SUD population cognitive flexibility as particularly impaired in patients. This is a comprehensive study, although there are some intrinsic limitations associated with the study design that impact the significance. Specific comments:
- The current matching criteria only included age and education, while many other factors that may well confound the study (social-economic status, drug history that may affect the inflammation, smoking and/or other lifestyle factors) were not balanced. This is a major limitation, and the conclusions may not be drawn appropriately without this information. Also, the current description of drug use (alcohol, methamphetamine, cannabinoids, inhalants, benzodiazepines) is broad and lacks detailed quantitative information. Please elaborate on drug history, if such information was collected, to allow better interpretation of the observed cognitive and inflammatory effects.
- Although CIBER provides visual clarity, it is still recommended to perform multivariate analysis for covariate effects, which provides more robust statistical significance.
- Figure 1: although there is a significant difference between SUD and control, it appears the statistical significance is driven by only one person. Please perform assessment of it being potentially an outlier. Nevertheless, given the small sample size, the authors should be cautious when interpreting the results. Also, there is no unit for IL-6 levels in the figure.
- Correlations coefficients between IL-6 and the assessed neuropsychological test performance are limited, despite being statistically significant (e.g., r values around 0.3-0.5), which limits the clinical significance. Thus, more cautious discussions are needed and the conclusion on correlations should be limited to exploratory rather than definitive.
- Abstract line 19-20: “pro-inflammatory cytokine interleukin-19 6 (IL-6) appears to contribute to these cognitive impairments significantly”. This sentence implies a causal relationship whereas the manuscript did not provide any evidence for the causal inference in the introduction.
- Abstract lines 18-20 would belong in the Background/Objectives and would be better to be moved after “Background/Objectives” in line 20.
- Line 66-69: it is important to distinguish whether the findings listed came from preclinical or clinical studies.
- In section 2.4, please add a reference regarding how the Wisconsin Card Sorting Test (WCST) was performed.
Author Response
Reviewer #2: The current study investigated the potential relationship between persistent inflammation and cognitive deficits in executive functions. Plasma IL-6 was used as a single marker for inflammation, and working memory, inhibitory control and attention, and cognitive flexibility were assessed using several neuropsychological tests during the withdrawal phase in male polydrug abusers compared to healthy controls. The study reported elevated IL-6 levels and executive dysfunction in the SUD population cognitive flexibility as particularly impaired in patients. This is a comprehensive study, although there are some intrinsic limitations associated with the study design that impact the significance. Specific comments:
Response: Thanks to Reviewer #2 for your insightful feedback and great comments to improve this manuscript.
Reviewer #2: The current matching criteria only included age and education, while many other factors that may well confound the study (social-economic status, drug history that may affect the inflammation, smoking and/or other lifestyle factors) were not balanced. This is a major limitation, and the conclusions may not be drawn appropriately without this information. Also, the current description of drug use (alcohol, methamphetamine, cannabinoids, inhalants, benzodiazepines) is broad and lacks detailed quantitative information. Please elaborate on drug history, if such information was collected, to allow better interpretation of the observed cognitive and inflammatory effects.
Response: We thank the reviewer for highlighting this important methodological limitation. Indeed, the matching criteria in our study were limited to age and education, and we acknowledge that other factors such as socioeconomic status, detailed drug use history, smoking, and additional lifestyle variables may act as confounding factors influencing both inflammatory markers and cognitive performance. While this limitation reduces the ability to draw firm causal conclusions, we believe the findings offer valuable preliminary evidence on the relationship between polydrug use, cognition, and inflammation, and can serve as a basis for future research with more comprehensive control of these variables.
Regarding the drug use history, although we documented the types of substances used, we did not collect detailed quantitative data (e.g., frequency, dosage, or duration), which indeed limits the precision with which the cognitive and inflammatory effects can be interpreted. This limitation has been acknowledged in the discussion section, and we suggest that future studies incorporate more comprehensive assessment tools to better characterize substance use profiles and associated contextual factors. Please see page 14, lines 559 to 571.
Reviewer #2: Although CIBER provides visual clarity, it is still recommended to perform multivariate analysis for covariate effects, which provides more robust statistical significance
Response: We thank the reviewer for this valuable suggestion. To complement the visual clarity provided by the CIBER analysis, an additional multivariate analysis was included in the manuscript (see paragraph 395). This analysis allows for a more robust assessment of covariate effects and yielded significant results (Pillai’s Trace = 0.77, F = 8.69, p < 0.001), supporting the validity of the model. Please see page 7, lines 348 to 354 and page 11, lines 401 to 402.
Reviewer #2: Figure 1: although there is a significant difference between SUD and control, it appears the statistical significance is driven by only one person. Please perform assessment of it being potentially an outlier. Nevertheless, given the small sample size, the authors should be cautious when interpreting the results. Also, there is no unit for IL-6 levels in the figure.
Response: We appreciate the reviewer’s important observation. Regarding the IL-6 levels, we verified that the value in question (37.47 pg/ml), although elevated, falls within the reference range reported in the literature for healthy individuals. According to Said et al. (2021), plasma IL-6 levels typically range from 0 to 43.5 pg/ml, with an average range between 0.13 and 30.46 pg/ml. Additionally, when compared to studies such as Lu et al. (2019), which report much higher levels in clinical populations, the observed value in our participants cannot be considered extreme. As noted by Merza and Mohammed (2021), outliers should not only be statistically deviant but also biologically implausible. After carefully inspecting the dataset, we conclude that although this IL-6 value is high, it does not meet the strict criteria to be classified as an outlier and provides relevant information within the context of the SUD group. Nevertheless, to ensure transparency, we performed a statistical outlier test (e.g., Tukey’s method), and the results have been added to the Results section. Furthermore, we have now included the appropriate unit of measurement (pg/ml) in Figure 1, as recommended, and have added a cautionary note in the Discussion section acknowledging the limitations of the small sample size when interpreting the findings. Please see page 4, lines 192 to 194.
Reviewer #2: Correlations coefficients between IL-6 and the assessed neuropsychological test performance are limited, despite being statistically significant (e.g., r values around 0.3-0.5), which limits the clinical significance. Thus, more cautious discussions are needed and the conclusion on correlations should be limited to exploratory rather than definitive.
Response: We thank the reviewer for this valuable comment. In response, we have revised the wording in the discussion section (lines 405–407), shifting from a definitive interpretation to a more cautious and exploratory tone. Please see pages, lines 430 to 442
Reviewer #2: Abstract line 19-20: “pro-inflammatory cytokine interleukin-19 6 (IL-6) appears to contribute to these cognitive impairments significantly”. This sentence implies a causal relationship whereas the manuscript did not provide any evidence for the causal inference in the introduction.
Response: We thank the reviewer for pointing out this issue and agree that the original wording in the abstract may indeed suggest a causal relationship, which is not directly supported by the evidence presented in the manuscript. In response, we have revised the sentence to reflect an associative rather than causal relationship. The updated text now reads: “the pro-inflammatory cytokine interleukin-6 (IL-6) has been associated with cognitive impairments in individuals with substance use disorders.”. Please see page 1, lines 18-20.
Reviewer #2: Abstract lines 18-20 would belong in the Background/Objectives and would be better to be moved after “Background/Objectives” in line 20.
Response: We thank the reviewer for this very constructive suggestion, and we apologize for our imprecise writing. In this revised version, we have moved the section Background/Objectives to lines 18 . Please see pages 1, lines 18-19.
Reviewer #2: Line 66-69: it is important to distinguish whether the findings listed came from preclinical or clinical studies.
Response: We apologize for our imprecise writing. In this version, we added clinical studies. Please see page 2, line 58.
Reviewer #2: In section 2.4, please add a reference regarding how the Wisconsin Card Sorting Test (WCST) was performed.
Response: Thanks for identifying this issue. In this revised version, we added the bibliographical citation . Please see page 4, line 157.
References:
Lu, J., Ma, S., Zhang, W.-Y., & Duan, J. (2019). Changes in peripheral blood inflammatory factors (TNF-α and IL-6) and intestinal flora in AIDS and HIV-positive individuals. Journal of Zhejiang University-SCIENCE B, 20(10), 793–802. https://doi.org/10.1631/jzus.B1900075
Merza, E. O., & Mohammed, N. J. (2021). Fast Ways to Detect Outliers. Journal of Techniques, 3(1), 66–73. https://doi.org/10.51173/jt.v3i1.287
Said, E. A., Al‐Reesi, I., Al‐Shizawi, N., Jaju, S., Al‐Balushi, M. S., Koh, C. Y., Al‐Jabri, A. A., & Jeyaseelan, L. (2021). Defining IL‐6 levels in healthy individuals: A meta‐analysis. Journal of Medical Virology, 93(6), 3915–3924. https://doi.org/10.1002/jmv.26654
Reviewer 3 Report
Comments and Suggestions for Authors
An overall a well constructed manuscript. I have provided a few comments and suggestions for improvement (See attached feedback).

Author Response
Reviewer #3: This investigation into inflammatory markers (IL-6 expression) and executive functioning within the withdrawal phase of male polydrug users is interesting and the use of CIBER methodology to accomplish this is a novel approach. The paper does well to explain the background, their methods, and presentation of their results. Overall, this investigation and associated paper will add to the growing body of nuanced findings related to addiction and recovery. I, however, suggest a few minor revisions including additional information, some word choices, and adjustments to figures be made and reviewed prior to acceptance for publication. Please see specific suggestions separated by section below.
Response: We thank the reviewer for their valuable comments and observations, allowing us to reflect on our introduction, methodology, results and discussion sections. We acknowledge
Reviewer #3: Line 21: remove the word “However” and capitalize the word “The” for the sentence.
Response: We thank the reviewer for her/his observation, we modified our text and added “the” rather than “however”. Please see page 2, line 20.
Reviewer #3 General: well structured and sets up the rest of the manuscript appropriately.
Response: We thank the reviewer for this kind point of view about our work
Reviewer #3: Line 88: A minor issue with word choice. “…subject paired by age, education and no story of drug consumption.” should read, “…subject paired by age, education and no history of drug consumption.” Additionally, this may need to be updated to include the word “problematic consumption behavior” as later in this section they state that the controls report “social consumption of alcohol” and “tobacco (one or two cigars per week).”
Response: We apologize for being somehow ambiguous in some parts of our manuscript. In this revised version, we modified. Please see page 2, lines 95-98, and page 3 line 106.
Reviewer #3: Line 97: Minor word choice “cups” should be “glasses”
Response: We thank the reviewer for her/his observation, we modified our text and added “glases”. Please see page 3, line 107.
Reviewer #3: Line 98: When discussing tobacco, only “cigars” was mentioned without discussing cigarettes or the growing trend of vaping both of which likely contain nicotine. Can the authors expand this point slightly or list this and self-report of this behavior more strongly in the limitations section at the end of the manuscript.
Response: We realize a broad examination of the state-of-the-art on this topic, but maybe we did have the ability to communicate it. Hence, we accept that our previous version might be confusing and lack crucial details that. Please see pages 3, lines 100-105 and pages 14, lines559-570
Reviewer #3: Line 147-169: The description of the shape of the data being non-parametric should be discussed prior to stating that the authors then utilized the non-parametric analysis of “Mann-Whitney U test.” Additionally, do the authors have enough power given their sample size to run this assessment?
Response: We apologize for not adding a better description; thus, in this revised version, we extended our original text with the following sentences to do a better analysis of these studies. Please see pages 4, lines 158 to 161
Reviewer #3: Line 169: Minor word choice, it should read “…expressed as 95% CIs (lower bound/upper bound).” or “CIs (lower limit/upper limit)”. This presentation should also be
consistent throughout the manuscript.
Response: We apologize for this, and we homogenized this term throughout the manuscript. Please see pages 4, line 183 and page 7, lines 313, 325.
Reviewer #3: Line 172: Authors state that their SUD group are “polydrug consumers” of methamphetamine, tobacco, and alcohol with some portion of their participants engaging in other substances. While stating that methamphetamine consumed with tobacco and alcohol is by definition polydrug use, it is not as typical as to if participants were using multiple illicit substances. While it may be fine in the body of the manuscript I ask the authors if it should be stated as such in the title of the manuscript.
Response: We thank the reviewer for highlighting this concern regarding the term “polydrug consumers” in our manuscript. According to internationally recognized definitions, the term “polydrug use” refers to the use of more than one drug or type of drug by an individual. This pattern encompasses both legal substances (alcohol and tobacco) and illegal substances (e.g. cannabis, cocaine, among others) (European Medicines agency, 2021). While the combination of methamphetamine with alcohol and tobacco may not reflect the most commonly studied pattern of multiple illicit drug use, it constitutes a valid form of polydrug use.
Reviewer #3: Line 180: Authors state, “We observed a worsening EF performance in the SUD group.” However, this cannot be stated unless they did multiple measures within the same group (i.e. repeated measures). The comparison group is the control group and you can use words like “lower” or “decreased compared to control” but not worsening because that would imply they were compared to themselves.
Response: We thank this observation and agree. We change the world “worsening” by the term “lower” with the aim of not generating misinterpretations. Please see page 5, line 196.
Reviewer #3: General: The term “significant” is a binary term meaning something is or isn’t significant. Authors can use terms like “approaching significance” but statements like “were at the limit of statistical significance” or “except by a limitrophe statistical significance…” and then interpreting as if it was significant is inappropriate. Additionally, you used the traditional metrics for this but made multiple comparisons without compensating by adjusting your significance threshold making it more likely to make a type 1 error. Thus, those that are near the threshold (p = 0.046) may be spurious.
Response: Thanks for pointing this out. We agree and in this revised version we avoid terms that could misinterpret the data with no statistical significance. We appreciate your insightful comment regarding the risk of type I error due to multiple comparisons. In response, we described in the methodology section (please see page 4, lines 167 to 171) an analysis based on the true effect. Our findings show that the possibility of detecting a type II error is less than 20% for effect sizes above 0.96 (r = 0.43). Meanwhile, to avoid a type I error, the effect size we obtained was greater than the minimum necessary value of 0.70 or more (r = 0.33). According to Lakens (2022), our findings suggest valid statistical criteria.
Additionally, we have now conducted a correction for multiple comparisons using Holm correction by analysis of variance and the results support our hypothesis. Details are shown on the next table. Thank you for helping us strengthen the rigor and interpretability of our results.
Test |
F |
p |
t |
pholm |
NBack right answer |
6.98 |
0.012 |
2.40 |
0.012 |
NBack reaction time |
4.56 |
0.040 |
2.14 |
0.040 |
Go-no-go right answer Go |
9.59 |
0.006 |
2.93 |
0.006 |
Go-no-go right answer no-go |
10.8 |
0.002 |
3.28 |
0.002 |
Go-no-go reaction time Go |
5.76 |
0.022 |
2.40 |
0.002 |
Go-no-go reaction time no-go |
10.5 |
0.003 |
3.25 |
0.003 |
WCST right answer |
23.4 |
< 0.001 |
34 |
<0.001 |
WCST reaction time |
7.42 |
0.010 |
2.72 |
0.010 |
WCST perseverative errors |
12.5 |
0.001 |
3.53 |
0.001 |
WCST non-perseverative errors |
6.01 |
0.020 |
2.45 |
0.020 |
Data are shown as sum of squares (F) with post hoc analysis with Holm correction. |
Reviewer #3: Line 296: Another instance of the presentation of confidence interval and lower and upper bounds. To be changed and consistent with the other instances in the manuscript.
Response:
Reviewer #3: Line 308: The term 95 % confidence interval has already been abbreviated earlier in the manuscript and does not need to be so once again here.
Response: Thanks for this suggestion. We remove the abbreviation and homogenize as previously suggested. Please see page 7, line 325.
Reviewer #3: Line 336: Figure 1 description states that the “red dot” symbolizes the median but since the traditional interpretation of the line in the box is the median the red dot is superfluous and can be removed.
Response: Thanks for the suggestion. In this revised version, we eliminate the red dot in the Figure 1. Please see page 8, lines 356.
Reviewer #3: Line 339-346: some issues with formatting for the Figure 2 description with some text being disassociated with the rest. Additionally, the authors should rephrase this slightly for clarity and space. I have made a suggestion here, “In the right panel the diamond colors indicate the direction with green colored suggesting a positive association compared to control and red having a negative association compared to control. Additionally, the strength of difference is indicated by color intensity with stronger colors representing stronger associations.”
Response: Thanks for the constructive suggestion. We corrected the formatting issues in the figure and add a better description in the figure´s legend. Please see page 8, lines 360.
Reviewer #3: Line 353: Suggest that the authors provide a distinguishing definition between the withdrawal phase that their participants are in (psychological withdrawal) vs those who
have only one or two days of abstinence, experiencing more physiological withdrawal.
Response: We thank the reviewer for this suggestion. In this study, none of the participants had one or two days of abstinence. As we mentioned in the methods section, all the participants in SUD group had approximately six months of abstinence. According to McGregor et al, (2005). The natural course of withdrawal in methamphetamine patients occurs in two phases, an acute one that involves the first 24hrs to 7 days following cessation. This phase comprises a linear decrease in the severity of the symptoms and a subacute phase where the symptoms remain relatively stable (the next two weeks). In the same sense, Zoric et al in (2010) reported similar findings and signaling that physiological symptoms decrease in the first weeks, but psychological symptoms as craving, could be present until six months after cessation. We add this bibliographical data to the discussion section to identify the withdrawal phase of our SUD groups. Please see page 14, lines 534 to 541
Reviewer #3: Line 361: Authors discuss a recent metanalysis related to their study but indicate that the primary drug that the current study population previously used (e.g., methamphetamine) was not included and was not indicated by the authors as to how this may impact the generalizability to their current study (I do think it is applicable and should be used but should inform the reader specifically as well).
Response: We apologize for the lack of clarity in our initial writing. Although the meta-analysis referenced does not specifically focus on methamphetamine as the primary substance of concern regarding IL-6, its indirect findings suggest that substance use, including methamphetamine, has been associated with an inflammatory profile characterized by elevated levels of several proinflammatory cytokines in peripheral blood, including IL-6. We have now clarified this point in the manuscript, and we think that although the meta-analysis referenced does not specifically focus on methamphetamine, its indirect findings are still relevant to our study population. We also emphasize the relevance of the meta-analysis to the broader context of substance use-related neuroinflammation. Please see page 12, lines 430 to 441.
Reviewer #3: Line 364-366: This sentence at least needs a comma, “… that in addictive behavior, exists biological…” or should be reformatted for comprehension.
Response: We apologize for this typo. We add a comma in the sentence. Please see page 12, lines 446
Reviewer #3: Line 372: authors state, “…suggests that a stressor stimulus…” are they indicating that the withdrawal phase acts as a stressor stimulus? If so, they should more clearly state this to help the reader make that connection.
Response: We thank for this constructive observation that helps us to better describe our ideas. We agree with the imprecise writing in these sentences, and we try to better describe this idea in the manuscript. Please see page 12 , lines 455-459
Reviewer #3: Line 376-377: Sentence is missing a period and should reference that this was done exclusively in males especially since the authors are juxtaposing it with a study only done
in women as per the prior line.
Response: We apologize for this typo. We add a period in the sentence and we clarify that this study was made on male patients. Please see page 13, line 477.
Reviewer #3: Line 382: Authors use an APA intext citation as opposed to AMA that they have used in the rest of the manuscript.
Response: Thanks for this observation. We change the citation to AMA style with the aim of homogenize the manuscript. Please see page 13, line 84
Reviewer #3: Line 423: Minor word agreement issue, “correct answers in the no-go condition
(commissions) and perseverative errors are very relevant cognitive parameters to beevaluated…”
Response: Thanks for this constructive suggestion to clarify our text. We change the sentence for a better grammatical use. Please see page 14, lines 526 to 528.
Reviewer #3: Line 425: suggest to change to “… design neuropsychology assessments and programs that will…” as the authors have noted that assessments not just programs may arise from this report.
Response: Thanks for helping us to improve our description of ideas. We agree with your suggestion and have changed the sentence as you mentioned. Please see page 14, lines 529 to 530
Reviewer #3: Line 436-450: This section is a bit odd in that it is just describing good research practice to find age-matched controls that are similar to your experimental group. While it is
appreciated that you must discuss that they are not different and can be well compared, it is not such that “the SUD group does not differ in age from the control group.” This is not a comparison to be made but more of a check that you did your job obtaining the control group appropriately. Again, I like the discussion that age can impact reaction times and other scores in the WCST much of this can be removed and this paragraph streamlined since the researchers did well to find a good control group that was age matched to their experimental sample.
Response: Thanks for this suggestion. We agree and modify the text and offer a brief paragraph summarizing the most important ideas of them. Please see page 14, lines 543 to 648
Reviewer #3: Line 467: Authors state that, “targeting these deficits…” but do not specify if this
targeting would be cognitively, pharmacologically, or both.
Response: Thanks for this inaccuracy in the writing. We specify cognitive deficits. Please see page 14, line 537 to 541.
Reviewer #3:Figure 1:There should be a value at the top of the y-axis especially if there is a data point that is near that range as is seen in the figure.
Response: We appreciate the reviewer for highlighting this concern regarding the IL-6 levels. According to Said et al. (2021), the normal range of IL-6 plasma levels in healthy people is 0 pg/ml to a maximum of 43.5 pg/ml, with a typical average range between 0.13 to 30.46 pg/ml. Our subject presented an IL-6 level of 37.47 pg/ml, while high, is below the maximum threshold of 43.5 pg/ml reported by Said et al. and is not as extreme as the values identified by Lu et al. (2019). Furthermore, as noted by Merza and Mohammed (2021), outlier values should be illogical in relation to the rest of the natural data. Therefore, we believe that this value is still relevant within the context of our study; it provides valuable information and does not meet the strict criteria for exclusion as an outlier despite its elevated level.
Reviewer #3 That top data point seems that it very well could be an outlier in your data and
does seem to be driving a lot of the effect for this small sample. What is the justification for not excluding something this far outside of the range of the other values (seems about 15 points higher than any of the other points)
Response: Thanks for this observation. We think that we also offer a justification of this point out in the previous response.
Reviewer #3 The titles on the X and Y axis should be increased in size and consistent in their
formatting (e.g., bolding).
Response: Thanks for this suggestion. We agree and change the size and homogenize the format. Please see page 8, lines 356
Reviewer #3: Providing the figure in a clearer format (e.g., above 300 DPI) would make the
image clearer and better convey the results presented by the authors.
Response: Thanks for this observation. We agree and modify the resolution of the image to the aim of offer more clarity in it. Please see page 8, lines 360
Reviewer #3: Not sure if this is due to the image quality or not but I am not detecting any
differences in the intensity of the color which is intended to represent the strength
of association.
Response: Thanks for this observation. With the suggestion made it in the previous comment, we improve the quality of the image and resolve this. Please see page 8, lines 360.
Reviewer #3: Suggest another color for either red or green as red-green color blindness is the
most common kind and will impact interpretability for some (about 1 in 12 men).
Response: Thank you for your kind comment about colour accessibility. We completely agree that it is essential to ensure the interpretability of figures for all readers. However, in this case, the colours were automatically assigned by the software and, unfortunately, the programme does not allow manual adjustments to the default colour scheme for this type of output. However, we provide a detailed description in the figure legend to clarify the meaning of each colour and minimize any possible confusion for the reader. Please se page 8, lines 365 to 368
Reviewer #3: A bit more separation of the titles on the x-axis to better delineate each panel
would be beneficial.
Response: Thanks for this suggestion. We modify this. Please see page 8, lines 460
Reviewer #3: The columns should be right or left aligned to organize the table a bit better. This
is particularly egregious under consumption type but should be with many of the
columns that appear to be center aligned causing movement in the table.
Response: Thanks for this suggestion. We agree and modify, and improve the quality of the format. Please see page 9, lines 374.
Reviewer #3: Consistent use of text font within the table is needed (the statistical tests are not
the same as the rest of the table).o The column header f (%) should be on the same line as the title “consumption type”
Response: Thanks for this suggestion. We agree and modify, and improve the quality of the format. Please see page 9, lines 374.
Reviewer #3: The IQR for “days with abstinence” is fairly large indicating a large amount of
variation and potential outliers in the data. Please make sure to reference this in
the manuscript (either results and/or in the limitations).
Response: Thanks for this suggestion. We agree and modify, and improve the quality of the format. Please see page 9, lines 374.
References:
European Medicines agency. (2021). http://www.ema.europa.eu/en/documents/annual-report/2021-annual-report-european-medicines-agency_en.pdf
Lakens, D. (2022). Sample Size Justification. Collabra: Psychology, 8(1). https://doi.org/10.1525/collabra.33267
Lu, J., Ma, S., Zhang, W.-Y., & Duan, J. (2019). Changes in peripheral blood inflammatory factors (TNF-α and IL-6) and intestinal flora in AIDS and HIV-positive individuals. Journal of Zhejiang University-SCIENCE B, 20(10), 793–802. https://doi.org/10.1631/jzus.B1900075
McGregor, C., Srisurapanont, M., Jittiwutikarn, J., Laobhripatr, S., Wongtan, T., & White, J. M. (2005). The nature, time course and severity of methamphetamine withdrawal. Addiction, 100(9), 1320–1329. https://doi.org/10.1111/j.1360-0443.2005.01160.x
Merza, E. O., & Mohammed, N. J. (2021). Fast Ways to Detect Outliers. Journal of Techniques, 3(1), 66–73. https://doi.org/10.51173/jt.v3i1.287
Said, E. A., Al‐Reesi, I., Al‐Shizawi, N., Jaju, S., Al‐Balushi, M. S., Koh, C. Y., Al‐Jabri, A. A., & Jeyaseelan, L. (2021). Defining IL‐6 levels in healthy individuals: A meta‐analysis. Journal of Medical Virology, 93(6), 3915–3924. https://doi.org/10.1002/jmv.26654
Zorick, T., Nestor, L., Miotto, K., Sugar, C., Hellemann, G., Scanlon, G., Rawson, R., & London, E. D. (2010). Withdrawal symptoms in abstinent methamphetamine‐dependent subjects. Addiction, 105(10), 1809–1818. https://doi.org/10.1111/j.1360-0443.2010.03066.x
Reviewer 4 Report
Comments and Suggestions for Authors
The manuscript employed abstinent substance use patients and healthy age-matched controls to measure IL-6 levels and executive function. The authors found that abstinent patients exhibited elevated IL-6 plasma levels and identified cognitive flexibility and inhibition as the most sensitive markers of executive dysfunction. These findings are clinically important and may provide insight for relapse prediction and treatment. The manuscript is well-written, and the data are mostly well-presented. There are some issues with the discussion and figures that are listed below:
- It is unclear whether participants in the SUD group had any drug use or relapse during the abstinence period or following the testing. If this information is available, it would be helpful to examine whether IL-6 levels or executive function performance are related to relapse risk or continued substance use. This is very important I think.
- The manuscript does not address whether elevated IL-6 reflects general systemic inflammation or a more specific neuroinflammatory response related to SUD. A brief discussion on this point would strengthen the interpretation of the findings.
- The reported correlations between IL-6 levels and cognitive performance are interesting. Including a figure to visually represent these associations would help communicate the results more clearly and emphasize their relevance.
- The use of abbreviations such as NBra, NBrt, etc., especially in Result section 3.5, makes the text difficult to follow. Although these abbreviations are defined at the beginning of the section, the dense use throughout reduces clarity and readability. Consider using full or more descriptive terms in the main text, and limiting abbreviations to figures and tables.
- Panel labels on x and y axes of Figure 1 are too small, please use the same font/font size for all figures throughout the manuscript.
Author Response
Reviewer #4: The manuscript employed abstinent substance use patients and healthy age-matched controls to measure IL-6 levels and executive function. The authors found that abstinent patients exhibited elevated IL-6 plasma levels and identified cognitive flexibility and inhibition as the most sensitive markers of executive dysfunction. These findings are clinically important and may provide insight for relapse prediction and treatment. The manuscript is well-written, and the data are mostly well-presented. There are some issues with the discussion and figures that are listed below:
Response: Thanks for the kind idea of this manuscript.
Reviewer #4: It is unclear whether participants in the SUD group had any drug use or relapse during the abstinence period or following the testing. If this information is available, it would be helpful to examine whether IL-6 levels or executive function performance are related to relapse risk or continued substance use. This is very important I think.
Response: Thanks for this suggestion. We agree, in this revised version, we modify the text in the material and methods and clarify that no SUD participants suffer from relapse during the experimental design. Please see page 3, lines 107 to 108
Reviewer #4: The manuscript does not address whether elevated IL-6 reflects general systemic inflammation or a more specific neuroinflammatory response related to SUD. A brief discussion on this point would strengthen the interpretation of the findings.
Response: Peripheral levels of cytokine interleukin 6 may contribute to elevated central IL-6 throught different mechanisms, such as active transporter across the blood-brain barrier and vagal nerve-mediated signaling. There is evidence were peripheral cytokines suchIL-6 can influence central nervous system function, inducing reactive gliosis, and promoting neuroinflammatory cascades that impair synaptic plasticity and cognitive performance. In people with substance use disorders, chronic systemic inflammation and cytokine dysregulation may exacerbate these effects, and maybe leading to greater cognitive impairments compared to non-using controls (Wilson et al., 2002)
Please see page 12, lines 430-442
Reviewer #4: The reported correlations between IL-6 levels and cognitive performance are interesting. Including a figure to visually represent these associations would help communicate the results more clearly and emphasize their relevance.
Response: Thanks for this suggestion, we added a new figure with the aim of improving the visual representation of this correlations. Please see page 10, line 383.
Reviewer #4: The use of abbreviations such as NBra, NBrt, etc., especially in Result section 3.5, makes the text difficult to follow. Although these abbreviations are defined at the beginning of the section, the dense use throughout reduces clarity and readability. Consider using full or more descriptive terms in the main text, and limiting abbreviations to figures and tables.
Response: Thanks to the reviewer for bringing this to our attention. In this revised version, we reduced the number of acronyms by 30% to improve the reading fluency. Please see pages lines 45, 56, 93, 186, 187, 189, 186, 193, 195-197, 197-199, 199, 205- 208, 209-211, 213, 214-216, 227, 232, 236, 241, 243, 249, 253-255, 258-261, 266-269, 271- 272, 279-283, 285, 289
Reviewer #4: Panel labels on x and y axes of Figure 1 are too small, please use the same font/font size for all figures throughout the manuscript.
Response: Thanks for this observation. We improve the format of the image to offer a better quality of the image. Please see page 8, lines 356.
Finally, we thank to all the reviewers for their time and for highlighting these important considerations; we believe these adjustments help strengthen our paper.
References:
Wilson, C. J., Finch, C. E., & Cohen, H. J. (2002). Cytokines and cognition--the case for a head-to-toe inflammatory paradigm. Journal of the American Geriatrics Society, 50(12), 2041–2056. https://doi.org/10.1046/j.1532-5415.2002.50619.x
Round 2
Reviewer 2 Report
Comments and Suggestions for Authors
The reviewer thanks the authors for the response, which addressed most of the comments. The reviewer recognizes some limitations that cannot be overcome at this point, and acknowledges the authors' effort in softening the language. There are a few additional comments:
- Figure 1: For some reason, the individual data points were removed in the box plot. Please include them to ensure transparency to readers, even if the test of outliers was performed.
- Please add to the method section how the multivariate analysis was conducted.
Author Response
Guadalajara, México. May 29th 2025
Re: Manuscript’s resubmission (ID: healthcare-3565164)
MDPI Healthcare Editorial Office
Grosspeteranlage 5, 4052 Basel, Switzerland
Dear Editorial Office Team,
We are grateful for the comments from your editorial board and your consideration of “Accepted after minor revisions” of our revised manuscript entitled “IL-6 expression and the Confidence Interval-Based Estimation of Relevance (CIBER) help identify persistent inflammation and cognitive parameters of executive dysfunction in the withdrawal phase of male polydrug abusers” for publication in the journal Healthcare.
In this version, we followed the reviewer’s last suggestions to improve our manuscript and have addressed the following:
- Improve the quality and edition of Figure 1.
- Better description of the method section.
Comments and responses:
Reviewer #2. The reviewer thanks the authors for the response, which addressed most of the comments. The reviewer recognizes some limitations that cannot be overcome at this point, and acknowledges the authors' effort in softening the language.
Response: Thanks for this kind point of view.
Reviewer #2. There are a few additional comments: Figure 1: For some reason, the individual data points were removed in the box plot. Please include them to ensure transparency to readers, even if the test of outliers was performed.
Response: Thanks for this suggestion and help to improve the quality and transparency of the data in this figure. We agree with you. In this version, we added the individual data points to the figure. Please see page 8, lines 362 to 367.
Reviewer #2. Please add to the method section how the multivariate analysis was conducted.
Response: Thanks for helping to identify this missing information. In this version, we added a better description of the multivariate analysis. Please see page 4, lines 184 to 190.
Sincerely,
Dr. Rocio Gonzalez-Castañeda on behalf of the authors.